

# Acute clomipramine exposure elicits dose-dependent surfacing behavior in adult zebrafish (*Danio rerio*)

Adeel Shafiq, Mercedes Andrade, Richanne Matthews, Alexandria Umbarger and Maureen L. Petrunich-Rutherford

Department of Psychology, Indiana University Northwest, Gary, IN, United States of America

## ABSTRACT

Chronic treatment with clomipramine, a tricyclic antidepressant drug, reduces symptoms of obsessive-compulsive disorder (OCD) and can influence the activity of the hypothalamic-pituitary-adrenal axis. However, little is known regarding the effects of acute clomipramine on the immediate expression of stress responses. Serotonergic drugs can elicit surfacing, a behavioral profile potentially related to toxicity in fish, although surfacing has not yet been observed after clomipramine exposure. The present study investigated the impact of acute exposure to clomipramine on basal and stress-induced behaviors in the novel tank test and cortisol levels in mixed-sex, wild-type, adult zebrafish (*Danio rerio*). The findings show clomipramine-exposed groups (regardless of stress exposure) spent much more time in the top of the novel tank and had significantly less overall motor activity in the behavioral task compared to the fish not exposed to the drug. Then, the dose-dependent effects of acute clomipramine on activity in the surface of the novel tank (top third of the top half) were investigated further. Clomipramine dose-dependently increased surface-dwelling and elicited a dose-dependent hypoactivity in overall motor behavior. There were no statistically significant differences in whole-body cortisol levels in either experiment. Like other serotonin-acting drugs, clomipramine strongly elicited surface-dwelling and depressed motor behavior in adult zebrafish. Additional testing is needed to elucidate whether surfacing represents a toxic state and how serotonin regulates surfacing.

## INTRODUCTION

Clomipramine is a tricyclic antidepressant with serotonin reuptake inhibitory actions and was the first drug to show clinical efficacy in reducing symptoms of obsessive-compulsive disorder (*Insel et al., 1983*). Today, clomipramine is reserved as a second-line treatment for patients that do not respond well to selective serotonin reuptake inhibitor (SSRI) pharmacotherapies (*Pittenger & Bloch, 2014*). Obsessive-compulsive disorder (OCD) has a lifetime prevalence of approximately 2.5%; this disorder is marked by intrusive thoughts and compulsions that, if left unfulfilled, can lead to intense bouts of anxiety (*Richter & Ramos, 2018*). Patients diagnosed with OCD also have significantly higher levels of

Corresponding author
Maureen L. Petrunich-Rutherford, mlpetrun@iu.edu

the stress-related hormone cortisol compared to healthy controls (*Sousa-Lima et al., 2019*). However, it is unclear whether clomipramine mitigates the dysfunction in the hypothalamic-pituitary-adrenal (HPA) axis observed in patients diagnosed with OCD (*Coryell et al., 1989*; *Gehris et al., 1990*). Currently, there are no treatments known that primarily or directly address the dysregulation of the HPA axis (*Menke, 2019*) that is observed in OCD and other stress-related conditions. Although there is little information available on the potential of acute clomipramine administration to modulate cortisol and behavioral responses simultaneously, clomipramine does have the potential to modulate HPA function (as has been observed with chronic treatment of animal models (*Frank, Gauthier & Bergeron, 2006*; *Fuchs et al., 1996*; *Prathiba, Kumar & Karanth, 1998*)) and elicit anxiolysis in human subjects (*Suetsugi et al., 1998*).

Animal models are frequently used to determine the anxiolytic potential and molecular mechanisms underlying the efficacy of drug compounds. Zebrafish (*Danio rerio*) have a high degree of morphological, physiological, neuroanatomic/neurochemical, and genetic homology to humans (*Kalueff, Stewart & Gerlai, 2014*; *Panula et al., 2010*), including a stress axis termed the hypothalamic-pituitary-interrenal (HPI) axis that is homologous to the mammalian HPA axis (*Wendelaar Bonga, 1997*). In addition, the results of neurobehavioral tests of exploration, anxiety, and locomotion in zebrafish generally parallel those traditionally seen in rodents (*Champagne et al., 2010*). Similar to other models, anxiety-like behavior in zebrafish is ascertained by exposure to novelty (*Fernandez & Gaspar, 2012*; *Hart et al., 2016*; *Kysil et al., 2017*), such as in the novel tank test (*Blaser & Rosemberg, 2012*; *Cachat et al., 2010*; *Raymond et al., 2012*; *Wong et al., 2010*). In the novel tank test, zebrafish initially display anxiety-like behaviors, such as bottom-dwelling, erratic movements, and immobility (*Maximino et al., 2010*). However, increased exploration of the top zone (*e.g.*, increased entries to top, increased distance in top, increased time in top) and increased overall mobility are observed across the test upon habituation. Different drug treatments and stressors modulate these innate behaviors of zebrafish in the novel tank, making zebrafish a useful model in drug screening (*Raymond et al., 2012*). Acute stressors elicit increased or prolonged anxiogenic effects on zebrafish behavior, while treatment with antidepressants, like fluoxetine, typically reduces overall anxiety-like behaviors and/or fosters an earlier habituation to the novel tank (*Egan et al., 2009*).

Anxiolytic top exploration in the novel tank test appears to be distinct from surfacing behavior, a potentially aberrant dwelling at the surface of the water. Surfacing is listed in the catalog of established zebrafish behavioral phenotypes and is often triggered by exposure to serotonin-modulating substances (*Kalueff et al., 2013*). Surfacing may be indicative of buoyancy dysregulation, such as with swim bladder disease (*Matthews, 2004*) and is consistently associated with increased time spent in the top zone, which would, at first glance, suggest that surfacing looks like anxiolysis in the novel tank test. However, surfacing may be indicative of a distinct state (*i.e.*, toxicity), since other behaviors measured in the novel tank test (*i.e.*, motor activity, freezing/immobility) do not reproduce the behavioral profile typically associated with anxiolysis. For instance, acute exposure to amitriptyline and imipramine, tricyclic antidepressants, are associated with increased top-dwelling but reduced motor behaviors (*e.g.*, decreased velocity and

decreased overall distance traveled) (*Demin et al., 2017*; *Kulikova et al., 2021*). Exposure to 3,4-methylenedioxymethamphetamine (MDMA) is marked by increased top exploration and impaired habituation (*Stewart et al., 2011*). Lysergic acid diethylamide (LSD) and mescaline, both psychedelics that target serotonin pathways, also increase top-dwelling behaviors in the absence of motor habituation in the novel tank test (*Grossman et al., 2010*; *Kyzar et al., 2012*). Hypolocomotion is typically associated with sedation, motor deficits, and/or abnormal neurological function (*Cachat et al., 2010*; *Kalueff et al., 2013*). More studies are needed to characterize surfacing behavior. Some research suggests surfacing behaviors in zebrafish could be a phenotypic sign of serotonin toxicity (*Demin et al., 2017*; *Stewart et al., 2013*).

Serotonin syndrome is associated with abnormal serotonergic activity and elevated levels of synaptic serotonin; the use of serotonin-modulating antidepressants, especially in combination with other serotonergic-acting substances, has been linked to serotonin syndrome in humans (*Foong et al., 2018*; *Francescangeli et al., 2019*; *Wang et al., 2016*). It is thought that drug-induced top-dwelling, reduced locomotion, and increased serotonin activity, may be characteristic of serotonin syndrome in zebrafish (*Demin et al., 2017*; *Stewart et al., 2013*). Thus, top-dwelling paired with hypolocomotion likely represents a profile distinct from anxiety-like behavior, although few, if any, studies have specifically characterized a surfacing profile in the novel tank test. With the increasing use of SSRIs, SNRIs, and other serotonergic-modulating compounds in the treatment of anxiety disorders, more research is needed to understand the unique nature of surfacing behaviors in zebrafish, how surfacing relates to anxiety and/or toxic states, and to understand how serotonergic drugs trigger surfacing behaviors in zebrafish.

To the best of our knowledge, the behavioral and neuroendocrine effects of clomipramine in adult zebrafish have not been studied. Thus, the original goal of the investigation was to determine whether acute exposure to clomipramine would mitigate the effects of an acute stressor on anxiety-like behavior and cortisol responses in zebrafish. Based on the results of study 1, we then modified our investigation to explore whether acute exposure to clomipramine would elicit dose-dependent surfacing behaviors. In the first study, anxiolytic behavior was *a priori* defined as increased exploration in the top (increased time spent and increased proportion of distance traveled in the top half of the novel tank). Based on the results of the first study, a ''surface'' zone was established (upper 1/6 of the tank) in the behavioral test for the second study. Behavioral parameters for surfacing behaviors were defined by increased surface dwelling (*e.g.*, increased time spent in surface zone, increased proportion of distance traveled in the surface, and fewer entries to surface). Based on previously published work using other tricyclic antidepressants (*Demin et al., 2017*; *Kulikova et al., 2021*), it was hypothesized that (1) clomipramine would prevent the behavioral and neuroendocrine effects of a 9-minute stressor and (2) clomipramine would dose-dependently increase surfacing behaviors and decrease motor activity (*e.g.*, reduced total distance traveled, reduced mean speed, and increased resting time). The results of this study will provide information regarding the potential of acute clomipramine to modulate behavior and modulate the stress axis. Furthermore, this study will provide additional information about top-dwelling *vs.* motor activity in clomipramine-treated fish

and provide additional support for the serotonergic modulation of surfacing behavior in fish species.

## MATERIALS & METHODS

### Subjects

Adult (approximately 6 months of age), naïve, mixed-sex, wild-type zebrafish (*Danio rerio*) ($N = 106$) were obtained from Carolina Biological Supply (Burlington, NC). Zebrafish were given at least ten days to acclimate to the laboratory environment before any experimental procedures were conducted (*Dhanasiri, Fernandes & Kiron, 2013*). The zebrafish were maintained on a recirculating system on a 14:10 light/dark cycle; the water temperature of the housing tanks (approximately 5 fish per liter in 1.8 L tanks) was maintained at $27 \pm 1$ °C. Other water parameters were measured twice weekly and were kept constant throughout the experiments: ammonia (0 ppm), nitrates (<20 ppm), nitrites (0 ppm), general hardness (~150 ppm), chlorine (0 ppm), and alkalinity (KH ~120 ppm). Zebrafish were fed flake food and dried ground brine shrimp once per day. All husbandry protocols were designed in accordance with published recommendations (*Matthews, Trevarrow & Matthews, 2002*; *National Research Council, 2011*; *Westerfield, 2000*) and were approved by the Indiana University School of Medicine-Northwest Institutional Animal Care and Use Committee (protocol IUSM-NW-IACUC-49). Animals were monitored for general health parameters before and during the experiment; no adverse events were observed.

### Materials

Clomipramine hydrochloride was obtained from Santa Cruz Biotechnology (Dallas, TX; catalog number sc-203898). Diethyl ether was purchased from Fisher Scientific (USA). Cortisol enzyme-linked immunosorbent assay kits were obtained from Salimetrics (State College, PA). Rectangular treatment tanks (1L crossing tanks with inserts removed, product #ZHCT100T) were purchased from Aquaneering (San Diego, CA, USA). BehaviorCloud software was used for all behavioral analyses (Columbus, OH, USA) (*BehaviorCloud, 2024*) and JASP (Amsterdam, Netherlands) (*JASP Team, 2024*) was used for all statistical analyses.

### Procedure

*Experiment 1: Clomipramine and stress.* Prior to data collection, a dosage test of clomipramine was done to determine the dose and time of exposure used in the experiment. Based on previous studies of zebrafish exposed to other tricyclic antidepressants (*Demin et al., 2017*; *Magno et al., 2015*; *Marcon et al., 2016*), trial doses of 0.5, 1.0, and 5.0 mg/L were chosen. The highest dose tested elicited aberrant behaviors possibly indicative of toxicity (*e.g.*, swimming upside down and vertically). The dose of 1.0 mg/L was chosen, as this was the highest dose tested that did not elicit aberrant swimming behaviors. On the day of the experiment, zebrafish home tanks were removed from the system and brought to the experiment room, which had lighting and temperature conditions equivalent to the housing room. Drug naïve fish (total $N = 51$) acclimated after the move for 30 min before procedures were initiated. Individual zebrafish subjects were randomly selected from the home tank and placed into either a clomipramine solution (1 mg/L system water)

or control (1 L system water) for twenty minutes. Then, half of the treated ($N = 12$) and half of the control zebrafish ($N = 13$) were individually exposed to a nine minute net acute stressor (*Ramsay et al., 2009a*). The nine-minute net stressor consisted of three minutes of air exposure in a clean net, three minutes in a clean 1L tank filled with system water, and three more minutes of air exposure. Non-stressed fish were taken from the clomipramine ($N = 13$) or control (system water) tanks ($N = 13$) and transferred into a new tank (1L fresh system water) for nine minutes. After treatments, each fish was transferred to the novel tank test, where behavior was recorded for six minutes (see Fig. 1A for diagram of experimental design). All treatment tanks were rinsed out and refilled with fresh solution for each subject; treatments were alternated so no two fish received the same treatment consecutively (*e.g.*, one fish was selected at random and was subjected to control conditions, then the next fish was selected randomly and was subjected to clomipramine). Fish were euthanized 15 min after being introduced to the novel tank.

*Experiment 2: Clomipramine dose-response.* Like Experiment 1, zebrafish home tanks were removed from the system and brought to the experimental room 30 min before treatments started. Subjects (total $N = 55$, 11 per each group) were randomly selected from the home tank and placed into either a clomipramine solution (0.125, 0.25, 0.5, or 1.0 mL/L system water) or control (1L system water) for twenty minutes. Treatments again were alternated so no two fish received the same treatment consecutively. Clomipramine solutions were prepared from a stock solution by the Principal Investigator; all treatments were masked to Research Assistants. After the 20-minute treatment, each fish was transferred to the novel tank test, where behavior was recorded for six minutes (see Fig. 1B for diagram of experimental design). All treatment tanks were rinsed out and refilled with fresh solution for each subject. Treatment conditions were unmasked after data analysis was completed. Fish were euthanized 10 min after being introduced to the novel tank.

*Novel tank test.* The motor activity and zone-dwelling behavior of each fish was measured using the novel tank test. The novel tank used in Experiment 1 and Experiment 2 was the same dimensions as the housing tanks (1.8 L), filled with clean system water to a depth of approximately 14 cm. The novel tank was placed on the lab bench next to white paper taped to the wall to obtain a better contrast of the fish *vs.* the background. Each individual fish was placed into the novel tank and was recorded for six minutes with a video camera connected to a tablet. The videos were then uploaded to the analysis software Behavior Cloud. In the software, top and bottom zones of the novel tank were drawn as the top 50% of the water column and the bottom 50% of the water column, respectively. In Experiment 2, the top zone was divided further; the top third of the top zone was considered the surface zone (see Fig. 2 for zone diagram). The video of each subject was analyzed *via* automated video tracking for the following measures: total distance traveled (cm), mean ambulatory speed (cm/s), total resting time (s), number of entries to the top or surface zone, total time in the top or surface zone (s), and total distance traveled in top or surface zone (cm). Distance traveled refers to the length of the pathway each subject traveled in the entire tank (total distance), only in the top zone (50% of the total water column), or only in

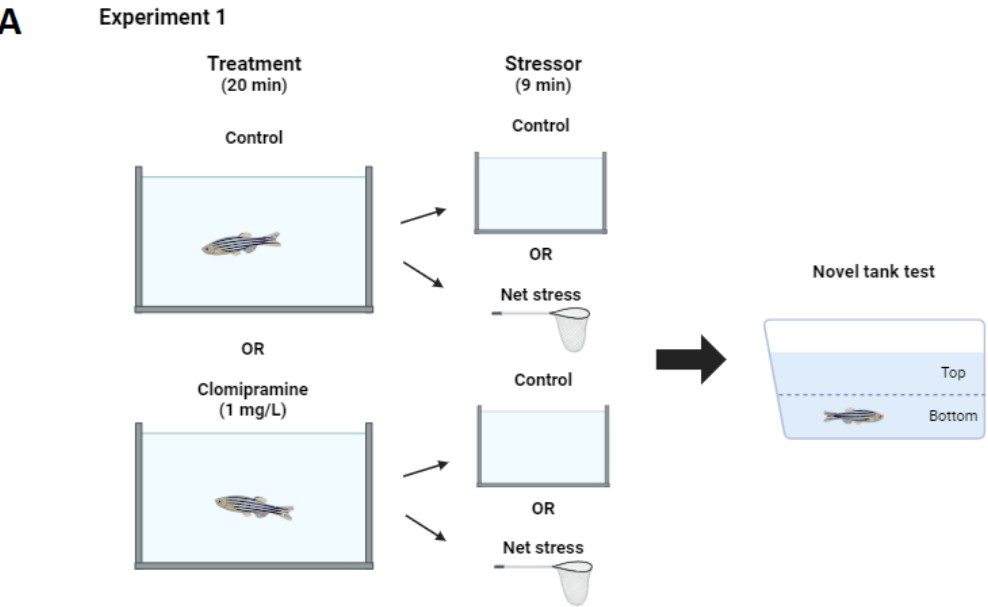

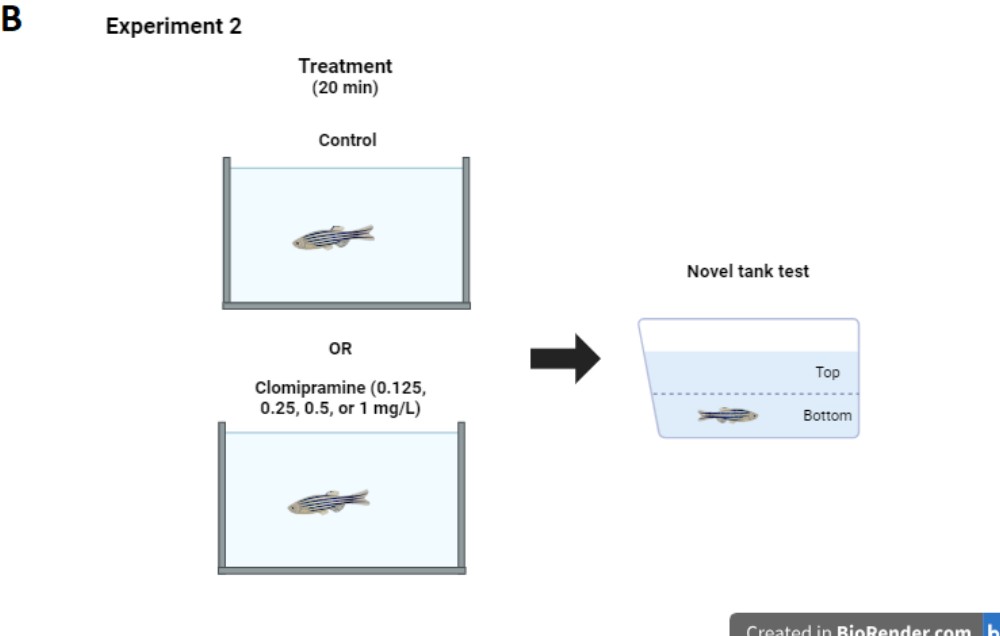

**Figure 1** **Experimental design.** Diagram of experiment design for Experiment 1 (A) and Experiment 2 (B). Created with BioRender.com.

the surface zone (1/3 of the top zone). The mean ambulatory speed (cm/s) was calculated by the software as the distance traveled divided by the seconds the subject is in motion (*i.e.,* not resting). Total resting time is determined by the software as the total number of seconds that the subject was not traveling. The number of entries refers to the number of

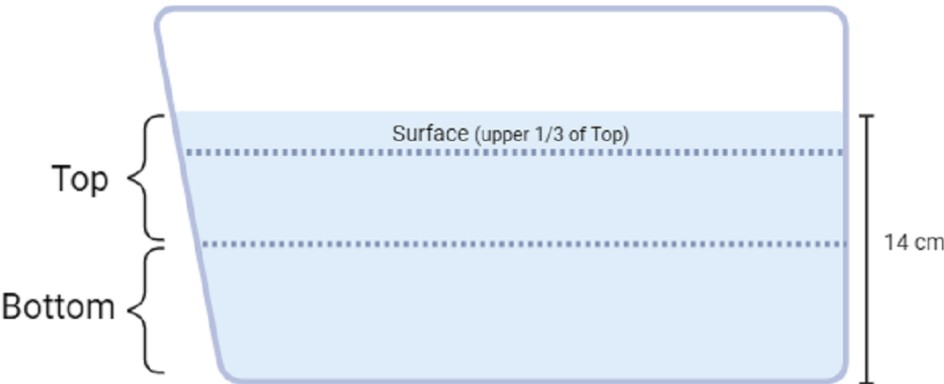

**Figure 2** **Novel tank test diagram.** Diagram of zone maps for the novel tank test. Created with BioRender.com.

times each subject moved into the top zone from the bottom zone. Total time in zone refers to the number of seconds the subject was resting or moving in the zone (top or surface). A proportion of the distance traveled in the top or surface was manually calculated by dividing the distance traveled in the top or surface zone by the total distance traveled. The proportion of distance traveled in the top or surface is represented as a percentage.

*Euthanasia.* Zebrafish were euthanized in approximately 30 ml of a 0.1% clove oil solution (1 ml clove oil/9 ml ethanol/990 ml system water) 15 min after starting the novel tank test for Experiment 1 and 10 min after starting the novel tank test for Experiment 2. The clove oil solution was utilized as a euthanasia medium, as it has been demonstrated to be less aversive than other agents, is fast acting, and does not impair cortisol responses (*Davis et al., 2015*; *Wong et al., 2014*). Zebrafish death was determined by immobility upon light tap with forceps as well as visual inspection of gill cessation, which usually occurred within seconds of being introduced to the solution. After one minute of remaining in the solution, death was determined, and each zebrafish was lightly dried with a Kimwipe and placed into a microcentrifuge tube on ice. Zebrafish whole-body samples were stored at −20 °C until cortisol extraction and assay.

*Whole-body cortisol assay.* Whole body cortisol levels were assessed using modifications to previously published procedures (*Aponte & Petrunich-Rutherford, 2019*; *Cachat et al., 2010*; *Canavello et al., 2011*). Briefly, zebrafish samples were thawed, weighed, cut into smaller pieces with a scalpel, and placed into a plastic test tube. Phosphate-buffered saline (PBS) solution (1 ml, 25 mM) was added to each tube; samples were homogenized for

60 s. Following homogenization, an extraction was performed with diethyl ether. Diethyl ether (5 ml) was added to each sample tube and vortexed for 60 s. Following the vortex, the tubes were placed into a refrigerated centrifuge (4 °C) for fifteen minutes at 2500 rpm. Once vortexed, the organic layer was removed from each sample and placed into a separate 20 ml glass test tube. The addition of ether, vortexing, centrifugation, and the removal of the organic layer was repeated two more times to maximize the cortisol extracted from each sample. Then, a stream of air was used to evaporate the ether in the test tubes, until only a yellow oil remained. Then, 1 ml of ice-cold PBS (25 mM) was added to each test tube. The test tubes were then covered with plastic wrap and stored at 4 °C overnight. In the morning following the extraction, the tubes were removed from storage and briefly vortexed. An enzyme-linked immunosorbent assay (ELISA) was performed as per manufacturer's instructions to determine the amount of cortisol in each sample. A standard curve was created; the mean absorbance of the duplicate wells for each sample without background absorbance was compared to the normal curve to determine the cortisol concentrations. Cortisol values were then normalized to the respective body weight for that sample before including in the statistical analysis.

*Statistical analysis.* A priori sample sizes were calculated with G\*Power (*Faul et al., 2007*), with a predicted moderate effect, $\alpha = 0.05$, and predicted power at least 80%. All analyses for this study were performed with the statistics software JASP. Values of $p < 0.05$ were considered statistically significant. When appropriate, Tukey *post-hoc* tests were used to determine differences between groups. For Experiment 1, a two-way analysis of variance (2-way ANOVA) with stress and drug as independent variables was conducted for each dependent variable (behavioral measures and cortisol values normalized to body weight). For Experiment 2, a one-way analysis of variance with drug dose as the independent variable was conducted for each dependent variable (behavioral measures and cortisol values normalized to body weight). For Experiment 2, Welch's correction was applied when Levene's test for equality of variances was statistically significant ($p < 0.05$). For Experiment 2, three cortisol values were out of range for the ELISA standard curve, resulting in each becoming an outlier for the respective group (>2 SD from the mean). Thus, these three values were removed from the overall cortisol analysis but the subjects' data remained in the behavioral analyses. All raw data have been provided in the Open Science Framework (*Petrunich-Rutherford, 2022*).

## RESULTS

### Experiment 1

*Top exploration.* Zebrafish exposed to clomipramine spent significantly more time in the top zone of the novel tank test (Fig. 3A), traveled proportionally more distance in the top zone (Fig. 3B), and entered the top zone fewer times (Fig. 3C) compared to control fish. However, in general, there was no consistent impact of the acute net stressor on any of these top exploration measures typically associated with anxiety-like behavior in control-treated fish. In clomipramine-treated fish, the effects of the drug on top exploration overshadowed any stress-induced alterations that may have been elicited by the stressor.

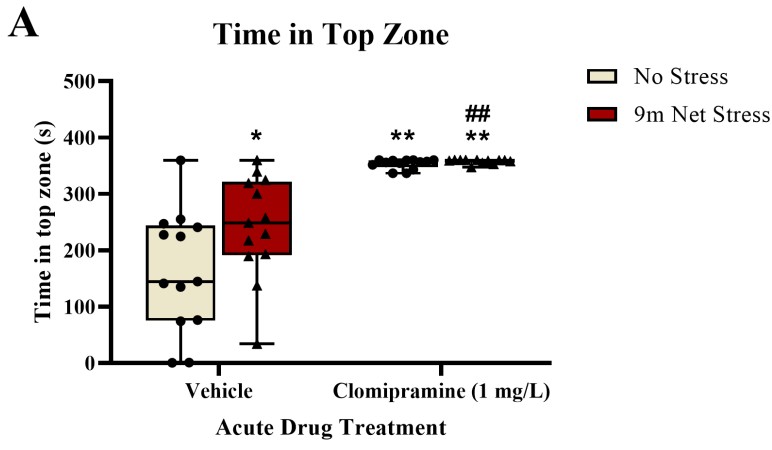

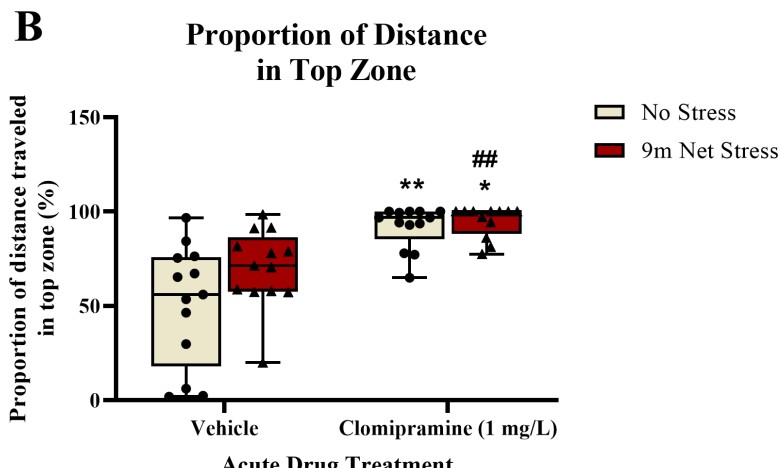

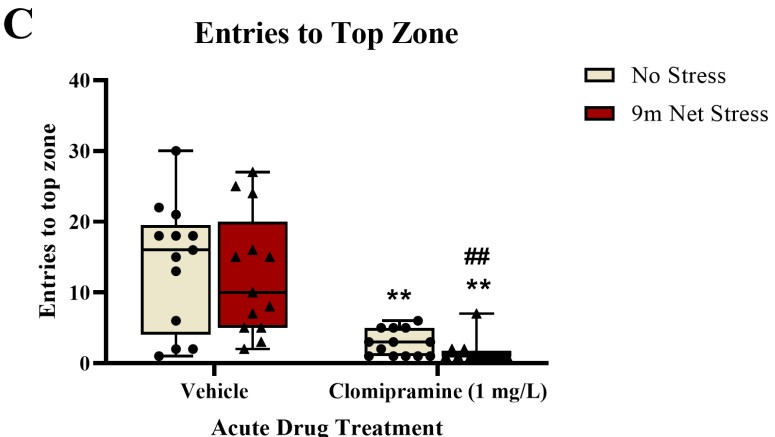

**Figure 3  The effects of clomipramine and stress exposure on top exploration measures in the novel tank test.** All data points are shown; the box indicates the 25th to 75th percentiles and whiskers indicate the minimum and maximum points. A 2-way ANOVA with Tukey *post-hoc* tests were utilized to analyze the data. Asterisks (*, **) indicate significant effect compared to respective vehicle control, $p < 0.05$ and 0.01, respectively; ## indicates significant effect compared to vehicle/no stress control, $p < 0.01$.

See Fig. 3A for the total time spent in the top zone of the novel tank test. There was a significant main effect of drug on total time in top ($F(1, 47) = 58.220$, $p < 0.001$) and a significant main effect of stress on total time in top ($F(1, 47) = 4.268$, $p = 0.044$). There was no significant interaction between drug and stress on total time in top ($F(1, 47) = 3.445$, $p = 0.070$).

For the proportion of distance traveled in the top zone of the novel tank (see Fig. 3B), there was a significant main effect of drug ($F(1, 47) = 32.790$, $p < 0.001$). There was no significant main effect of stress on proportion of distance traveled in top ($F(1, 47) = 3.734$, $p = 0.059$). There was no significant interaction between drug and stress on proportion of distance traveled in top ($F(1, 47) = 2.058$, $p = 0.158$).

For the number of entries to the top zone of the novel tank (see Fig. 3C), there was a significant main effect of drug on top zone entries ($F(1, 47) = 37.709$, $p < 0.001$). There was no significant main effect of stress on top zone entries ($F(1, 47) = 0.578$, $p = 0.451$). There was no significant interaction between drug and stress on top zone entries ($F(1, 47) = 0.010$, $p = 0.920$).

These data suggest that acute exposure to clomipramine elicits a surfacing behavior in adult zebrafish, where subjects quickly enter and spend most of the trial time in the top zone without exiting the zone.

*Motor activity.* Zebrafish exposed to clomipramine traveled a significantly shorter distance (Fig. 4A), had a decreased mean ambulatory speed (an effect that was statistically significant, Fig. 4B), and spent significantly more time resting (Fig. 4C) in the novel tank test compared to control fish. There was a significant main effect of drug on distance traveled ($F(1, 47) = 29.297$, $p < 0.001$), mean ambulatory speed ($F(1, 47) = 13.107$, $p < 0.001$), and resting time ($F(1, 47) = 16.619$, $p = 0.001$). However, there was no statistically significant main effect of acute stressor on total distance traveled ($F(1, 47) = 1.167$, $p = 0.285$), mean ambulatory speed ($F(1, 47) = 3.134$, $p = 0.083$), or resting time ($F(1, 47) = 0.409$, $p = 0.525$). There was no significant interaction between clomipramine treatment and stress exposure on total distance traveled ($F(1, 47) = 1.906$, $p = 0.174$), mean ambulatory speed ($F(1, 47) = 0.927$, $p = 0.341$), or resting time ($F(1, 47) = 1.643$, $p = 0.206$). Thus, acute exposure to clomipramine resulted in a depressant-like effect on motor activity in adult zebrafish in this study.

*Whole-body cortisol.* As illustrated in Fig. 5, there was a trend towards a decrease in whole-body cortisol levels in subjects exposed to clomipramine compared to control. However, this effect was not statistically significant ($F(1, 47) = 2.847$, $p = 0.098$). There was also no effect of stress on cortisol ($F(1, 47) = 1.517$, $p = 0.224$), nor was there an interaction between drug and stress on cortisol ($F(1, 47) = 0.221$, $p = 0.640$). This suggests that neither clomipramine nor acute stressor altered whole-body levels of cortisol in adult zebrafish, at least at this time point after drug and stress exposure.

## Experiment 2
*Surface dwelling.* Zebrafish acutely exposed to clomipramine demonstrated a dose-dependent increase in time in the surface zone of the novel tank test (Fig. 6A), traveled

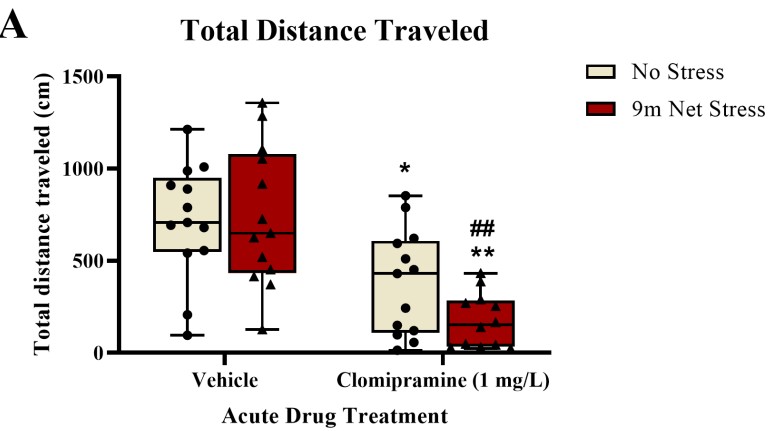

**A** **Total Distance Traveled**

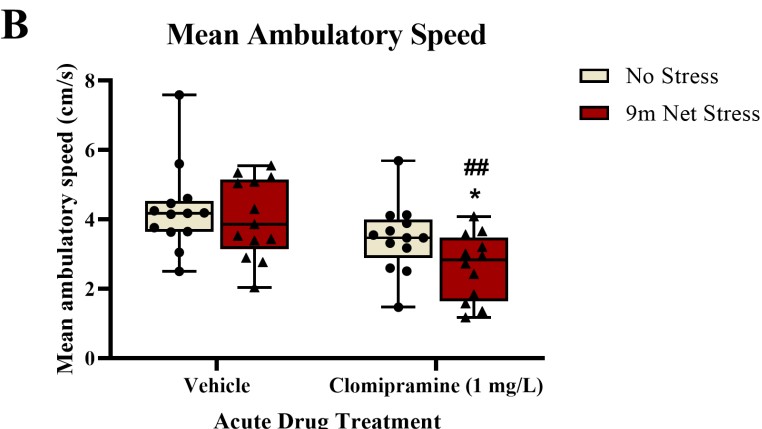

**B** **Mean Ambulatory Speed**

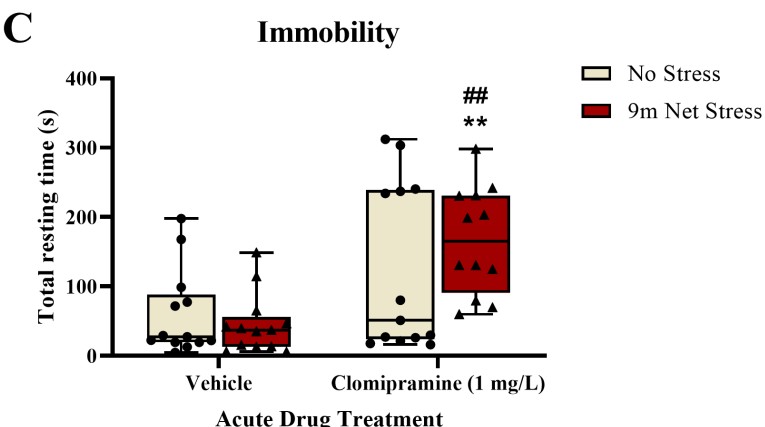

**C** **Immobility**

**Figure 4** **The effects of clomipramine and stress exposure on general motor activity in the novel tank test.** All data points are shown; the box indicates the 25th to 75th percentiles and whiskers indicate the minimum and maximum points. A 2-way ANOVA with Tukey *post-hoc* tests were utilized to analyze the data. Asterisks (*, **) indicate significant effect compared to respective vehicle control, $p < 0.05$ and 0.01, respectively; ## indicates significant effect compared to vehicle/no stress control, $p < 0.01$.

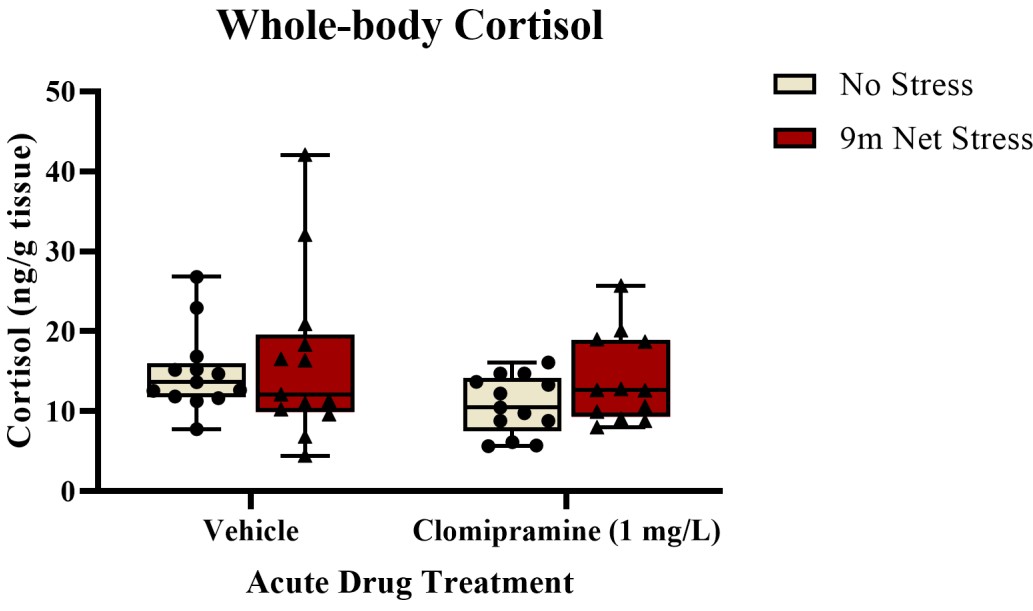

**Figure 5** **The effects of clomipramine and stress exposure on whole-body cortisol levels in zebrafish.**
All data points are shown; the box indicates the 25th to 75th percentiles and whiskers indicate the minimum and maximum points. A two-way ANOVA was utilized to analyze the data.

proportionally more distance in the surface zone (Fig. 6B), and generally entered the surface zone fewer times (Fig. 6C) compared to control fish.

There was a significant main effect of drug on total time in the surface zone ($F(4, 24.031) = 564.488$, $p < 0.001$), see Fig. 6A. For the proportion of distance traveled in the surface zone of the novel tank (see Fig. 6B), there was a significant main effect of drug ($F(4, 23.527) = 118.957$, $p < 0.001$). There was a significant main effect of drug on surface zone entries ($F(4, 23.740) = 5.010$, $p = 0.005$), see Fig. 6C.

These data demonstrate that acute exposure to clomipramine dose-dependently elicits surface-dwelling behaviors in adult zebrafish. Especially with higher doses, fish are traveling to the surface and are spending more time there, spending a proportionally higher amount of the distance traveled in the surface, and entering fewer times.

*Motor activity.* Zebrafish exposed to clomipramine showed a dose-dependent decrease in total distance traveled (Fig. 7A) and mean ambulatory speed (Fig. 7B), and clomipramine-exposed fish generally spent more time resting (Fig. 7C) in the novel tank test compared to control fish. There was a significant main effect of drug on distance traveled ($F(4, 24.714) = 15.572$, $p < 0.001$), mean ambulatory speed ($F(4, 24.173) = 6.882$, $p < 0.001$), and resting time ($F(4, 22.738) = 3.039$, $p = 0.038$). Like the results from Experiment 1, acute exposure to clomipramine dose-dependently increased depressant-like motor activity in adult zebrafish.

*Whole-body cortisol.* As illustrated in Fig. 8, basal levels of cortisol were inconsistently affected by acute exposure to clomipramine. Fish exposed to the dose of 0.25 mg/L

**A**

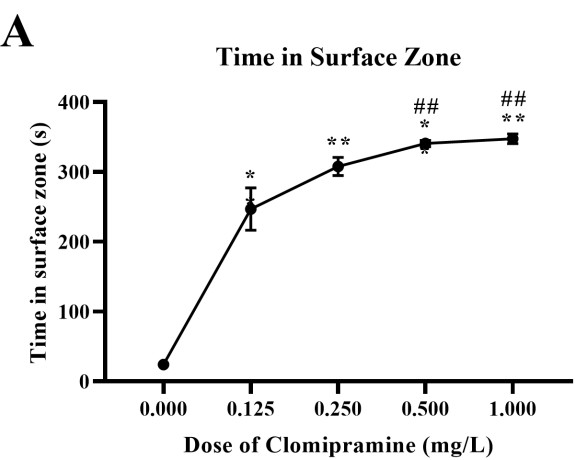

**B**

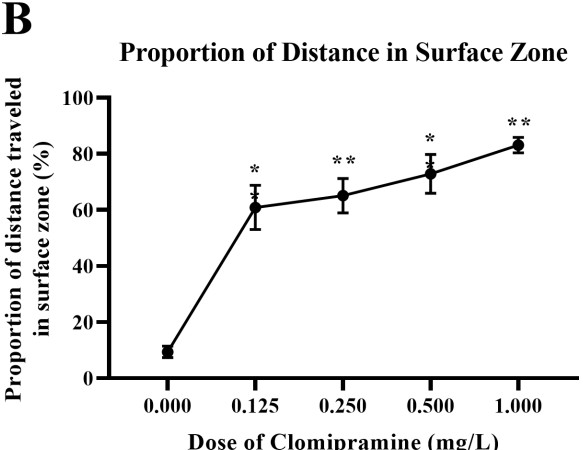

**C**

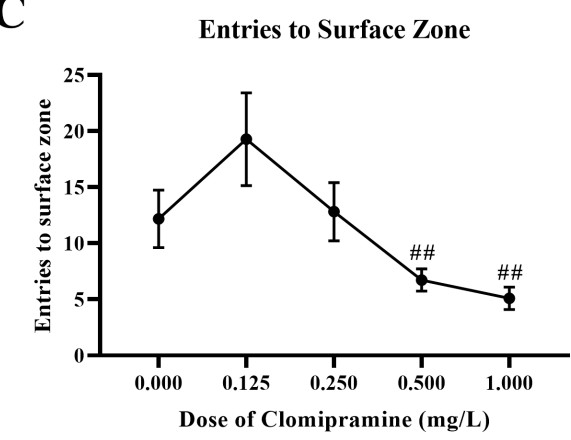

**Figure 6** **Dose-dependent effects of clomipramine exposure on top dwelling measures in the novel tank test.** Error bars indicate standard error of the mean (SEM). A 1-way ANOVA with Tukey *post-hoc* tests were utilized to analyze the data. Asterisks (\*, \*\*) indicate significant effect compared to vehicle (0.0 mg/L) control, $p < 0.01$; # indicates significant effect compared to 0.125 mg/L clomipramine, $p < 0.05$.

**A**

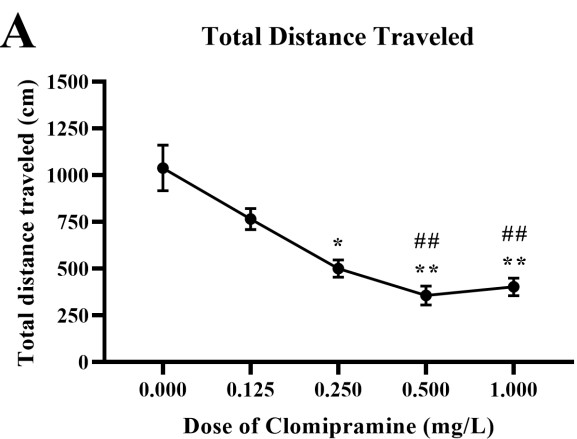

**B**

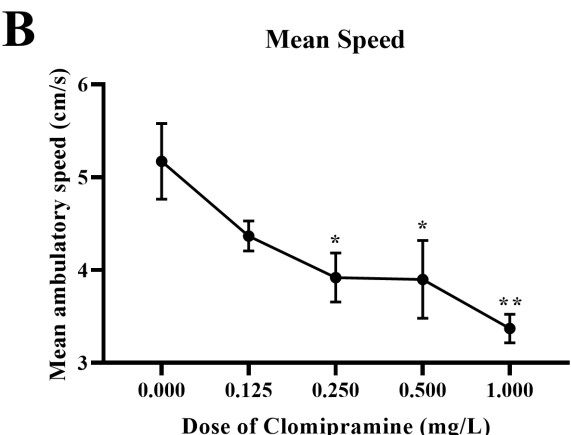

**C**

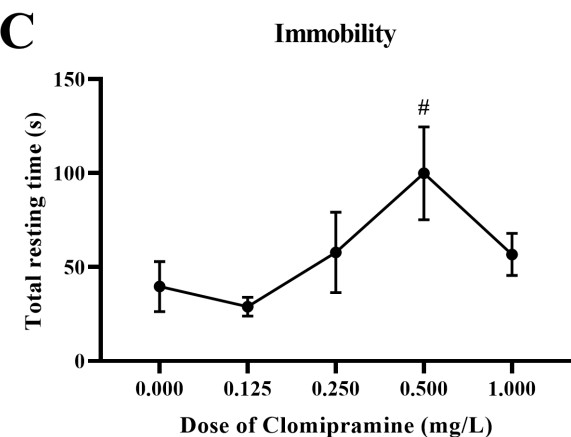

**Figure 7** **Dose-dependent effects of clomipramine exposure on general motor activity in the novel tank test.** Error bars indicate standard error of the mean (SEM). A 1-way ANOVA with Tukey *post-hoc* tests were utilized to analyze the data. Asterisks (*, **) indicate significant effect compared to vehicle (0.0 mg/L) control, $p < 0.05$ and 0.01, respectively; #, ## indicates significant effect compared to 0.125 mg/L clomipramine, $p < 0.05$ and 0.01, respectively.

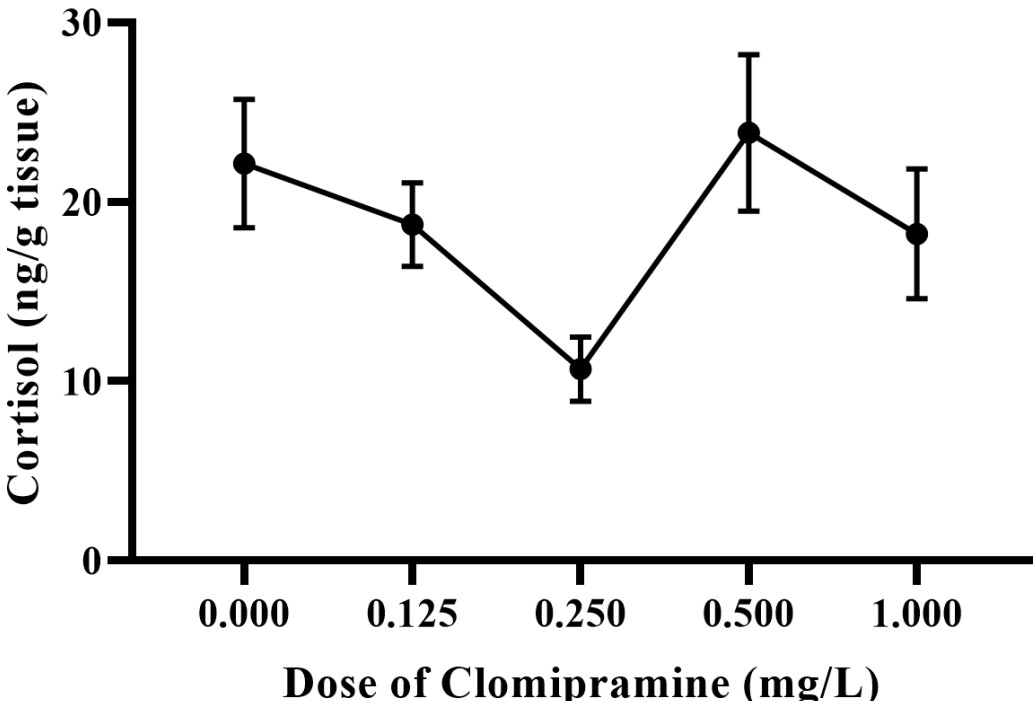

**Figure 8** Dose-dependent effects of clomipramine exposure on whole-body cortisol levels in zebrafish. Error bars indicate standard error of the mean (SEM). A one-way ANOVA was utilized to analyze the data.

clomipramine had the lowest levels of cortisol of all doses tested (approximately 50% of control). After Welch's homogeneity correction, the effect of drug on whole-body cortisol levels was statistically significant ($F(4, 22.773) = 3.854$, $p = 0.016$). Although less conclusive than the behavioral data, clomipramine may have the potential to modulate baseline levels of cortisol in adult zebrafish at certain doses or at certain time points after drug exposure.

## DISCUSSION

To the best of our knowledge, the present study is the first to investigate the effects of acute clomipramine exposure on behavioral and neuroendocrine responses in adult zebrafish. The original hypothesis was that (1) clomipramine would prevent the behavioral and neuroendocrine effects of a 9-minute stressor and (2) clomipramine would dose-dependently increase surfacing behaviors and decrease motor activity (*e.g.*, reduced total distance traveled, reduced mean speed, and increased resting time). Clomipramine exposure (20 min) consistently and dose-dependently increased surfacing-dwelling and decreased overall activity in the novel tank. In Experiment 1, fish exposed to clomipramine (regardless of stress condition) entered the top zone fewer times but remained in the top zone near the

surface of the water for more time and traveled most of the total distance in the top zone of the novel tank. Thus, in Experiment 2, we focused on measuring surface zone activity to determine if subjects were dwelling in the surface rather than exploring the top zone more generally as has been done in other studies. Indeed, clomipramine dose-dependently increased surface dwelling in the novel tank. In both experiments, clomipramine decreased total distance traveled and mean speed; clomipramine also slightly increased resting time but did not consistently alter whole-body cortisol levels. The behavioral effects observed in our study are consistent with other studies utilizing acute exposure to other tricyclic antidepressants (*Demin et al., 2017*; *Kulikova et al., 2021*), namely, increased top-dwelling and depressed motor activity; however, this study is novel in that behavior in the surface zone was specifically examined rather than top zone activity more generally.

Although alterations in brain chemistry elicited by acute clomipramine exposure were not investigated in the present study, given the mechanism of action of clomipramine, changes in behavioral activity following acute clomipramine treatment would indicate modulation of the central serotonergic system. Clomipramine, a tricyclic antidepressant, primarily functions as a serotonin reuptake inhibitor (*Hollander et al., 2000*; *Wilson & Tripp, 2021*), although it has demonstrated affinity for the norepinephrine transporter and some neurotransmitter receptor subtypes (*Millan et al., 2001*). Thus, dose-dependent increases in surfacing behaviors elicited by clomipramine may reflect an increasingly elevated serotonergic tone. Distinguishing surface-dwelling (surfacing), top exploration (anxiolytic), and motor activities elicited by different serotonergic-acting substances may help further the understanding of the distinct contribution of individual serotonergic pathways involved with different behaviors (*Ohmura et al., 2020*). For example, the current study and others examining tricyclic antidepressants indicate that acute and chronic treatment increases surfacing and reduces motor activity. However, other serotonin-modulating substances, such as MDMA, LSD, and mescaline, can elicit surfacing behaviors in the absence of overall motor changes (*Grossman et al., 2010*; *Kyzar et al., 2012*; *Stewart et al., 2011*). It is possible that serotonergic psychedelics and tricyclic antidepressants target the same pathways that mediate surfacing behavior, but only tricyclics simultaneously depress motor activity.

Whether surfacing represents a toxic state remains to be fully elucidated. Given that many of the drugs that elicit surfacing behavior have serotonergic mechanisms of action, it has been suggested that surfacing behaviors are a phenotypic marker of serotonin toxicity. In humans and animal studies, serotonin toxicity is characterized by abnormally elevated serotonin levels. In animal models, serotonin syndrome-like states have been associated with alterations in motor activity, such as increased hypertonicity, head waving, altered locomotion, and increased anxiety (*Kalueff, LaPorte & Murphy, 2008*; *Stewart et al., 2013*). Although the characteristics of a serotonin syndrome-like state have not been clearly defined in zebrafish, it has been suggested that increased hypolocomotion, increased surfacing, and increased serotonin may indicate signs of serotonin toxicity (*Stewart et al., 2013*). This profile could be distinct from those related to anxiety states and measuring the activity of the surface of the novel tank (top third of the top half) may be more useful for assessing surfacing compared to measuring top zone activity more generally. Increased exploration of

the top zone of the novel tank test typically indicates anxiolysis while impaired habituation and increased bottom-dwelling typically represents anxiety-like behavior (*Cachat et al., 2010*). Thus, the dose-dependent increase in surfacing behaviors following clomipramine treatment may be a marker of a serotonin syndrome-like state.

With regards to stress, increased whole-body cortisol levels generally rise in parallel with drug- and stress-induced anxiety-like behavior (*Egan et al., 2009*; *Rosa et al., 2018*). Thus, we expected that cortisol levels would also increase with surfacing behavior if surfacing represented toxicity in adult zebrafish. We also expected that the 9-minute acute stressor used in Experiment 1 would increase whole-body cortisol levels when fish. However, the 9-minute net stress did not significantly alter cortisol levels 15 min post-stress as was observed in a previous study (*Ramsay et al., 2009a*). In that study, all fish in the stress exposure group were stressed and euthanized as a group, while here we exposed the fish to the net stressor individually. It is possible that the proximity to conspecifics in a net is more stressful than being isolated in a net. It is also possible that an environmental stressor was present in our laboratory setting that raised the baseline cortisol so that any stress-induced cortisol elicited by the 9-minute stressor would not be noticeable. At least one study illustrated an elevation of cortisol elicited by husbandry stressors associated with housing and handling (*Ramsay et al., 2009b*). However, in the current study, the stressors also did not produce anxiety-like behavior in fish exposed to 9-minutes of net stress; control (water-treated)/stress-exposed fish spent slightly more time in the top zone of the novel tank but did not differ from control fish on any other behavioral metric. Clomipramine-treated zebrafish entered the top zone almost immediately and remained there for the duration of the test, regardless of stress exposure. Thus, it appears that the 9-minute net stress protocol was not sufficient to elicit prototypic stress responses in the current study. Future studies examining the impact of clomipramine treatment on acute stress responses should use a more severe stressor, such as acute restraint stress (*Ghisleni et al., 2012*; *Lucas Luz et al., 2020*), that consistently increases cortisol and alters top exploration in individual zebrafish. Alternatively, other physiological measures of stress could be used in future studies to compliment whole-body cortisol levels, such as brain levels of corticotropin-releasing hormone (CRH) mRNA (*Ghisleni et al., 2012*) or c-fos levels (*Ariyasiri et al., 2021*). Future research could also incorporate additional behavioral assessments to further understand how surfacing relates to anxiety in zebrafish, such as the light-dark test or open field test. Utilizing alternative behavioral assessments may facilitate a more complete understanding of surfacing as a behavioral profile distinct from anxiety states.

There were a few limitations to the current study. First, a mixed-sex group was used to investigate the effects of clomipramine on behavior. Sex differences are observed with regards to the efficacy of antidepressants in both clinical and preclinical models (*LeGates, Kvarta & Thompson, 2019*). There were significant sex by treatment effects observed on anxiety-like behavior elicited by seven days of clomipramine treatment in rodents (*Jimenez et al., 2018*), indicating that the possibility that acute clomipramine exposure could elicit different effects in male and female zebrafish. Sex differences have been observed in response to different stressors and pharmacological manipulations in adult and juvenile zebrafish

(*Cui et al., 2021*; *Genario et al., 2020*; *Nielsen et al., 2019*; *Rambo et al., 2017*; *Thompson, Shvartsburd & Vijayan, 2022*). Thus, it may be necessary for future studies to investigate sex differences in response to clomipramine treatment to better elucidate the specific effects of the drug on stress-related behavioral and neuroendocrine measures.

Given the putative role of different serotonergic pathways in surfacing behaviors in zebrafish, future work should investigate alterations of serotonin levels elicited by clomipramine and other serotonin-acting drugs. Acute exposure to amitriptyline, a tricyclic antidepressant, dose-dependently increased serotonin turnover at the level of the whole brain in zebrafish (*Demin et al., 2017*), but decreased serotonin turnover when administered chronically (*Meshalkina et al., 2018*). Levels of serotonin, norepinephrine, and other brain chemicals were also altered with seven days of amitriptyline exposure, alterations that did not always return to baseline after a 21-day recovery period (*Qiu et al., 2022*). Additionally, antagonizing specific subtypes of serotonin receptors could provide additional information about which receptors are involved with surfacing. In a previous study, antagonizing 5-HT$_{1A}$, 5-HT$_2$, and 5-HT$_3$ receptors in zebrafish elicited anxiety-like behaviors while antagonizing 5-HT$_{1B/D}$ receptors elicited anxiolytic behavior. It is notable that these actions were observed in the absence of significantly altered motor activity (*Nowicki et al., 2014*). Thus, measuring brain levels of serotonin, metabolites, and/or other neurotransmitters will also help clarify whether surfacing is elicited by excess levels of serotonin. Elucidating the contributions of individual receptor subtypes would also be valuable.

Future research should evaluate potential differences in the acute and chronic effects of clomipramine treatment on stress-related biochemical markers and behavioral responses. Although limited research has investigated the behavioral effects of chronic antidepressants in adult zebrafish, there appear to be different behavioral effects depending on drug class. It appears that chronic fluoxetine treatment (an SSRI) does not increase surfacing behavior but does decrease anxiety-like behaviors in the novel tank test depending on the exposure time. Seven days of fluoxetine exposure did not significantly alter motor or top measures on its own (although did prevent chronic stress-induced anxiety-like behavior) (*Marcon et al., 2016*), but 14 days of fluoxetine significantly increased top-dwelling behaviors in novel tank test without depressing motor activity (*Egan et al., 2009*). On the other hand, chronic treatment (14 days) with amitriptyline, a tricyclic antidepressant, increased surfacing behavior (increased top dwelling alongside decreased motor activity) (*Meshalkina et al., 2018*). This evidence suggests that surfacing behaviors again could selectively be elicited by certain drug classes. This evidence also suggests that zebrafish do not habituate to chronic tricyclic antidepressant exposure, as surfacing behaviors are still observed with two weeks of amitriptyline treatment (*Meshalkina et al., 2018*), although further studies are necessary to determine whether chronic exposure to clomipramine or other tricyclic antidepressants elicit similar effects.

## CONCLUSIONS

The present study demonstrated that acute clomipramine exposure modulates zebrafish behavior; namely, acute clomipramine increases surfacing (top-dwelling) behaviors and

depresses motor activity in adult, mixed-sex zebrafish. Acute clomipramine does not appear to modulate hypothalamic-pituitary-interrenal (HPI) activity, suggesting that drug exposure is not inherently stressful (at least at this dose range and treatment time). Given that clomipramine putatively blocks serotonin reuptake in the synapse (*Wilson & Tripp, 2021*), the current study adds to the body of literature that suggests that serotonin modulation is involved with surfacing behaviors in zebrafish. Additional work will help to clarify the behavioral profiles (surfacing, anxiety, and/or toxic states) associated with antidepressant exposure in adult zebrafish.

## ACKNOWLEDGEMENTS

The authors would like to thank Emma (DiPasquo) Meekma, Aidan Aubuchon, Cara Blanford, Tonimarie Cribari, Shelby Ford, Amaya Hostetler, Jessica Nguyen, Mariam Shatat, and Elizabeth Weer for their technical assistance with some aspects of these studies. Any opinions, findings, and conclusions or recommendations expressed in this material are those of the author(s) and do not necessarily reflect the views of the National Science Foundation

### Funding

This work was supported by Indiana University Northwest internal funding opportunities. Further, support for undergraduate researchers was provided by the National Science Foundation under Grant No. 1618-408. The funders had no role in study design, data collection and analysis, decision to publish, or preparation of the manuscript.

### Grant Disclosures

The following grant information was disclosed by the authors:
Indiana University Northwest internal funding opportunities.
National Science Foundation: 1618-408.

### Competing Interests

The authors declare there are no competing interests.

### Author Contributions

- Adeel Shafiq conceived and designed the experiments, performed the experiments, analyzed the data, authored or reviewed drafts of the article, and approved the final draft.
- Mercedes Andrade conceived and designed the experiments, performed the experiments, analyzed the data, authored or reviewed drafts of the article, and approved the final draft.
- Richanne Matthews conceived and designed the experiments, performed the experiments, analyzed the data, authored or reviewed drafts of the article, and approved the final draft.

- Alexandria Umbarger conceived and designed the experiments, performed the experiments, analyzed the data, authored or reviewed drafts of the article, and approved the final draft.
- Maureen L. Petrunich-Rutherford conceived and designed the experiments, performed the experiments, analyzed the data, prepared figures and/or tables, authored or reviewed drafts of the article, and approved the final draft.

### Animal Ethics

The following information was supplied relating to ethical approvals (*i.e.*, approving body and any reference numbers):

The Indiana University School of Medicine - Northwest Institutional Animal Care and Use Committee provided approval and oversight of this work.

### Data Availability

The data is available in the Open Science Framework: Petrunich-Rutherford, Maureen L. 2023. ''Acute Clomipramine Exposure in Adult Zebrafish, 2019-2023''. OSF. April 7. DOI: 10.17605/OSF.IO/XHSBF.

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
