# Peer review of "Acute clomipramine exposure elicits dose-dependent surfacing behavior in adult zebrafish (Danio rerio)"

_PeerJ, doi:10.7717/peerj.17803_

## Round 0.1 · original submission · Major Revisions

I have now received the reviewers' comments on your manuscript. They have suggested some major revisions to your manuscript. Therefore, I invite you to respond to the reviewers' comments and revise your manuscript.

Reviewer 1 ·

Basic reporting

Overall, the manuscript is well-written, the methodological approach is robust, and the authors have done commendable work. However, while the data is intriguing, there are certain aspects that require further clarification to enhance the reader's comprehension.

Experimental design

Despite the study being well conducted, I would like to highlight some points for improvement by the authors:

1) Line 36-37: When individuals are considering pharmacotherapy as the primary treatment for obsessive-compulsive disorder (OCD), the preference is to start treatment with a selective serotonin reuptake inhibitor (SSRI) rather than alternative antidepressants such as clomipramine. Please see the references below:
Pigott TA, Pato MT, Bernstein SE, Grover GN, Hill JL, Tolliver TJ, Murphy DL. Controlled comparisons of clomipramine and fluoxetine in the treatment of obsessive-compulsive disorder. Behavioral and biological results. Arch Gen Psychiatry. 1990 Oct;47(10):926-32. doi: 10.1001/archpsyc.1990.01810220042005
López-Ibor JJ Jr, Saiz J, Cottraux J, Note I, Viñas R, Bourgeois M, Hernández M, Gómez-Pérez JC. Double-blind comparison of fluoxetine versus clomipramine in the treatment of obsessive compulsive disorder. Eur Neuropsychopharmacol. 1996 May;6(2):111-8. doi: 10.1016/0924-977x(95)00071-v
Zohar J, Judge R. Paroxetine versus clomipramine in the treatment of obsessive-compulsive disorder. OCD Paroxetine Study Investigators. Br J Psychiatry. 1996 Oct;169(4):468-74. doi: 10.1192/bjp.169.4.468
Pigott TA, Seay SM. A review of the efficacy of selective serotonin reuptake inhibitors in obsessive-compulsive disorder. J Clin Psychiatry. 1999 Feb;60(2):101-6. doi: 10.4088/jcp.v60n0206
Soomro GM, Altman D, Rajagopal S, Oakley-Browne M. Selective serotonin re-uptake inhibitors (SSRIs) versus placebo for obsessive compulsive disorder (OCD). Cochrane Database Syst Rev. 2008 Jan 23;2008(1):CD001765. doi: 10.1002/14651858.CD001765.pub3

2) The authors did not provide details regarding the randomization procedure employed to assign animals to their respective intervention groups. Moreover, it is essential to clarify the method used for random allocation to these treatment groups and ascertain whether the experimenters and data analysts were blinded to the treatment conditions. This crucial information should be incorporated into the Materials and Methods section of the manuscript.

3) I believe that in addition to monitoring temperature, it is imperative to control other physical-chemical properties of water, such as pH and conductivity, throughout the experiments. Unfortunately, these crucial details are lacking from the methods section.

4) The fact that acute stress does not induce an increase in cortisol is inconsistent with similar studies previously published. Is there a possibility that the animals in the control group were already under basal stress from other circumstances, and consequently, the measured cortisol levels were at the ceiling (ceiling-effect)?

5) I found the description of the criteria for outlier detection and exclusion very interesting. However, I disagree with the authors in the sense that if a specific individual was excluded for one outcome, they should also be excluded from the entire experiment. I suggest reviewing this approach.

6) Is there any possibility that the observed behavior with clomipramine is due to a toxic effect or some peripheral modulation independent of the HPI axis?

Validity of the findings

The data analysis is accurate and effectively presented in the manuscript.

Reviewer 2 ·

Basic reporting

This manuscript is well written with clear language throughout. It was easy to read and would certainly relay well to a general audience. Authors are to be commended for this. Extensive referencing is also worth noting as a strength of this manuscript.
Introduction
Introduction is well-written; however, changes can be made to the overall structure to improve writing flow as well as clarity.
For instance, from lines 46-48, there is mention of animal model systems being used. These would be more appropriate towards the end of the paragraph leading into the next paragraph which focuses on animal models. Lines 49-51 would be ideally placed before this.
Identifying the subtle differences between zebrafish anxiolytic top exploration and surfacing behaviour is a crucial component in this study and the authors have an entire paragraph dedicated to fleshing out these two phenotypes. However, in lines 74-66, it is suggested that surfacing is consistently associated with increased time spent in the top zone, and this would be indicative of anxiety-like behaviour. However, time spent in the top is commonly associated with less anxiety. Egan et al, 2009 as well as many other papers also postulate this. If the authors can add some clarity to this statement, it would go a long way. Another aspect worth commenting on is how surfacing can be associated with physical impairments or inability to maintain buoyancy, as opposed to just being evoked by selected neuroactive drugs. Overall, this paragraph can also be shortened (for instance, Lines 81-87 can potentially be cut).
The final paragraph of the introduction can also be shortened significantly to focus on the question at hand, how this will be done and what the authors hypothesize. For instance, lines 108-118 seem more appropriate in the methods section of the manuscript, in the Procedure section prior to describing Experiments 1 and 2. This would add some context into why there were two different experiments performed.
Materials and Methods
Considering the sequential study design, preparing a figure which summarizes the experiment would be beneficial to helping the reader visualize the different facets of the experiment. It would also supplement the text very effectively. For examples, see Fontana et al. 2021, Duarte et al. 2019, Quadros et al. 2021.
References
Fontana, B.D., Alnassar, N. & Parker, M.O. The zebrafish (Danio rerio) anxiety test battery: comparison of behavioral responses in the novel tank diving and light–dark tasks following exposure to anxiogenic and anxiolytic compounds. Psychopharmacology 239, 287–296 (2022). https://doi.org/10.1007/s00213-021-05990-w
Duarte T, Fontana BD, Müller TE, Bertoncello KT, Canzian J, Rosemberg DB. Nicotine prevents anxiety-like behavioral responses in zebrafish. Prog Neuropsychopharmacol Biol Psychiatry. 2019 Aug 30;94:109655. doi: 10.1016/j.pnpbp.2019.109655. Epub 2019 May 18. PMID: 31112733.
Egan, R. J., Bergner, C. L., Hart, P. C., Cachat, J. M., Canavello, P. R., Elegante, M. F., ... & Kalueff, A. V. (2009). Understanding behavioral and physiological phenotypes of stress and anxiety in zebrafish. Behavioural brain research, 205(1), 38-44.
Quadros, V. A., Rosa, L. V., Costa, F. V., Koakoski, G., Barcellos, L. J., & Rosemberg, D. B. (2021). Predictable chronic stress modulates behavioral and neuroendocrine phenotypes of zebrafish: Influence of two homotypic stressors on stress-mediated responses. Comparative Biochemistry and Physiology Part C: Toxicology & Pharmacology, 247, 109030.

Experimental design

The rationale for this experiment is well thought out and aims to address a gap in the understanding of behavioural and cortisol responses associated with clomipramine exposure in zebrafish.
Statistical analysis is sound however it seems the authors pooled males and females together when analysing results. Sex differences have been shown to be important biological variable and there have been repeated calls for it to be included in analyses (Zajitschek et al. 2020). Furthermore, there is evidence that psychiatric disorders such as OCD also have a basis to looked at differently for men and women (Franceschini and Fattore, 2021; DeCasien et al. 2022); and also differently for male and female zebrafish in anxiety-based tests (Dos Santos et al. 2021). With this in mind, the analysis can be strengthened by adding sex as an independent variable in ANOVA tests and interpreting results as such.
It is mentioned that zebrafish were adults, can the authors please clarify the specific age of the animals when they were tested?
It is also mentioned that animals were monitored for general health parameters. Were the zebrafish bred for maintenance purposes? Females can develop health issues such as being eggbound if they are not breeding enough. If so, how long post-breeding were the animals tested? Some clarity on this may be beneficial (perhaps in a supplementary section).
Can the authors provide a diagram for the methods section which showcases the tank used as well as the divisions made in the tank?
In lines 158-159, authors state that treatments were staggered. Can this be elaborated on to provide an overview of how treatments were administered to avoid bias?
References
Franceschini, A., & Fattore, L. (2021). Gender-specific approach in psychiatric diseases: Because sex matters. European journal of pharmacology, 896, 173895.
DeCasien, A.R., Guma, E., Liu, S. et al. Sex differences in the human brain: a roadmap for more careful analysis and interpretation of a biological reality. Biol Sex Differ 13, 43 (2022). https://doi.org/10.1186/s13293-022-00448-w
Dos Santos, B. E., Giacomini, A. C., Marcon, L., Demin, K. A., Strekalova, T., de Abreu, M. S., & Kalueff, A. V. (2021). Sex differences shape zebrafish performance in a battery of anxiety tests and in response to acute scopolamine treatment. Neuroscience Letters, 759, 135993.
Zajitschek, S. R., Zajitschek, F., Bonduriansky, R., Brooks, R. C., Cornwell, W., Falster, D. S., ... & Nakagawa, S. (2020). Sexual dimorphism in trait variability and its eco-evolutionary and statistical implications. elife, 9, e63170.

Validity of the findings

The authors present their results clearly and concisely with relevant figures that are labelled, and colour coded for ease of interpretation. The authors also do a very good job at stating the purpose as well novelty of their study in the first paragraph of the discussion and re-iterate their hypotheses and results to establish the foundation of the discussion. The discussion is also well structured with detailed referencing, touching upon limitations and a succinct conclusion. Some minor comments below.
The authors found found that 9-minute net stress did not significantly alter cortisol levels 15 minutes post-stress. This is also addressed by Philippe at al. who found that cortisol levels were still elevated after 30 min and dropped to resting level after 60 min. This could potentially indicate that the timing is crucial in detecting cortisol levels and different timings may be required.
Other studies have also shown that using taller tanks would also be beneficial in detecting subtle differences in zebrafish behaviour (Anwer et al. 2021).
If possible, future studies may also aim to look at methods at extracting cortisol while keeping animals alive to perform repeated measures and take into account repeatability measures as well as mixed effects models to account for individual variances. Such an approach can help flesh out nuances associated with zebrafish behaviour in response to clomipramine.
References
Anwer, H., Mason, D., Zajitschek, S., Noble, D. W., Hesselson, D., Morris, M. J., ... & Nakagawa, S. (2021). An efficient new assay for measuring zebrafish anxiety: Tall tanks that better characterize between-individual differences. Journal of Neuroscience Methods, 356, 109138.
Philippe, C., Vergauwen, L., Huyghe, K., De Boeck, G., & Knapen, D. (2023). Chronic handling stress in zebrafish Danio rerio husbandry. Journal of Fish Biology, 103(2), 367–377. https://doi.org/10.1111/jfb.15453

Additional comments

No comment

Reviewer 3 ·

Basic reporting

We would like to extend our congratulations to the authors for their well-written paper and the clear presentation of the majority of the methodology. The chosen topic is interesting and falls clearly within the scope of the journal.

While certain points have been highlighted throughout the form, the majority of feedback can be found in the attached file.

Upon a general review, the introduction is both relevant and well written, although slightly lengthy. The paper does, however, lack some important information which hinders the ability to replicate the study accurately, while also overselling the potential impact of the results. The references to existing literature are pertinent, although there are a few missing sources that could have made a significant difference in the discussion.

In terms of structure, the paper is satisfactory. The figures provided are mostly relevant, although the absence of raw data contradicts the prevailing trend of open science.

Experimental design

Original research within the designated scope.

The research question is adequately defined, although the authors appear to diverge somewhat on the potential toxicity and serotonin impact, regardless of their analysis.

There is a requirement for clearer definition and description of the behaviors being analyzed, as the ethogram (specific set of observed behaviors) is not provided (e.g., freezing, immobilizing, resting). More precise definitions of these behaviors would facilitate a more thorough understanding of the implications of the results.

Technically, the paper demonstrates sound methodology and adheres to ethical standards.

Validity of the findings

The significance of the impact appears to be overstated. However, the novelty of the findings is acceptable. Replication of the study could be achieved by providing additional information, which is requested in the attached file. Conducting a study that examines the effects of clomipramine on cortisol levels would greatly benefit the scientific community by enhancing our understanding of serotonin modulation when using antidepressants and further advancing the zebrafish model.

While the conclusions are clearly expressed, they tend to stretch beyond what the results can actually substantiate. It is recommended to avoid relying on non-significant trends when drawing conclusions.

Additional comments

see attached file

Annotated reviews are not available for download in order to protect the identity of reviewers who chose to remain anonymous.

---

## Round 0.2 · accepted · Accept

An external reviewer and I are satisfied with the changes that have been made to the manuscript.

Reviewer 1 ·

Basic reporting

The authors have done excellent work, and the manuscript has been significantly improved. I recommend this manuscript for publication in PeerJ.

Experimental design

No comments.

Validity of the findings

No comments.

Additional comments

No comments.